# Post-decision processing in primate prefrontal cortex influences subsequent choices on an auditory decision-making task

Joji Tsunada[1,2], Yale Cohen[1,3,4†]*, Joshua I Gold[3†]

[1]Department of Otorhinolaryngology, University of Pennsylvania, Philadelphia, United States; [2]Department of Veterinary Medicine, Faculty of Agriculture, Iwate University, Morioka, Japan; [3]Department of Neuroscience, University of Pennsylvania, Philadelphia, United States; [4]Department of Bioengineering, University of Pennsylvania, Philadelphia, United States

**Abstract** Perceptual decisions do not occur in isolation but instead reflect ongoing evaluation and adjustment processes that can affect future decisions. However, the neuronal substrates of these across-decision processes are not well understood, particularly for auditory decisions. We measured and manipulated the activity of choice-selective neurons in the ventrolateral prefrontal cortex (vlPFC) while monkeys made decisions about the frequency content of noisy auditory stimuli. As the decision was being formed, vlPFC activity was not modulated strongly by the task. However, after decision commitment, vlPFC population activity encoded the sensory evidence, choice, and outcome of the current trial and predicted subject-specific choice biases on the subsequent trial. Consistent with these patterns of neuronal activity, electrical microstimulation in vlPFC tended to affect the subsequent, but not current, decision. Thus, distributed post-commitment representations of graded decision-related information in prefrontal cortex can play a causal role in evaluating past decisions and biasing subsequent ones.
DOI: https://doi.org/10.7554/eLife.46770.001

*For correspondence:
ycohen@pennmedicine.upenn.
edu

[†]These authors contributed equally to this work

## Introduction

Perceptual decision-making is a deliberative process that produces a categorical judgment regarding the presence, identity, and other features of a sensory stimulus (*Gold and Shadlen, 2007*). This deliberation often requires resolving potentially ambiguous interpretations of the current sensory stimulus with expectations that can be learned by evaluating prior decisions and their outcomes (*Fecteau and Munoz, 2003*; *Gold and Stocker, 2017*). This learning process can result in sequential effects on a subject's choices and response times (RTs) when they participate in psychophysical tasks that require one decision after another (*Gold et al., 2008*; *Marcos et al., 2013*; *Akaishi et al., 2014*; *Hwang et al., 2017*; *Akrami et al., 2018*; *Busse et al., 2011*; *Fischer and Whitney, 2014*; *Abrahamyan et al., 2016*; *Luu and Stocker, 2018*). Because sequential effects can be present even when a task is designed to generate independent trial-by-trial choices and after extensive training, these effects may represent fundamental, ongoing processes that evaluate and adjust decisions to account for changes in the sensory environment, reward contingencies, and other factors (*Fan et al., 2018*; *Gold and Stocker, 2017*). Although neuronal substrates of these sequential effects have been identified in several brain regions (*Barraclough et al., 2004*; *Ding and Gold, 2010*; *Ding and Gold, 2012a*; *Carnevale et al., 2012*; *Fecteau and Munoz, 2003*; *Marcos et al., 2013*; *Hwang et al., 2017*; *Akrami et al., 2018*; *Gold et al., 2008*; *Akaishi et al., 2014*; *St John-Saaltink et al., 2016*;

*Lueckmann et al., 2018*; *Histed et al., 2009*), the exact nature of the signals that the brain uses to support these effects is still not well understood.

In a previous study, we demonstrated that neurons in middle-lateral (ML) and anterolateral (AL) belt regions of the auditory cortex encode key features of the sensory evidence needed to solve an auditory-decision task (*Tsunada et al., 2016*). This task required monkeys to report whether a tone-burst sequence contained more low- or high- frequency tone bursts. We found that both AL and ML neurons were modulated by the frequency content of the tone-burst sequence. Moreover, in AL, but not ML, neuronal activity was weakly related to choice, and microstimulation biased the monkeys' choices. These findings suggest a more direct role for AL in the decision process than for ML. They are also consistent with the idea that AL provides evidence for the decision but leave open the question of where and how in the brain is this evidence interpreted and combined with other information to form the decision (*Gold and Shadlen, 2007*).

The goal of the present study was to identify a role for the ventrolateral prefrontal cortex (vlPFC) in forming this auditory decision. We targeted vlPFC because it receives direct and indirect projections from AL and is situated at the apex of the ventral auditory pathway, which is commonly thought to mediate auditory perception and decision-making (*Romanski et al., 1999*; *Hackett et al., 1999*; *Rauschecker and Tian, 2000*; *Russ et al., 2008b*; *Tsunada et al., 2016*; *Bizley and Cohen, 2013*). Here we show that, contrary to our initial expectations, vlPFC neurons do not appear to encode information relevant to forming the current decision. Instead, vlPFC population activity can encode rich, graded information about the just-completed decision process, including the strength of the sensory evidence, the resulting choice, and whether or not it was correct. These signals, which were apparent from just after the decision was formed until after feedback was received, were closely and causally related to the subsequent decision in a manner that matched each monkey's idiosyncratic choice strategy. Together, these results imply a role for the vlPFC in the ongoing evaluation and adjustment of auditory decisions.

## Results

We recorded and manipulated vlPFC spiking activity in two monkeys while they reported whether a noisy auditory stimulus contained more low- or high-frequency tone bursts (*Figure 1a,b*). A primary benefit of this task is that we could control the strength of the sensory evidence (the fraction of low or high tone bursts in a given stimulus; that is coherence) and relate that evidence to the monkeys' choices and to vlPFC activity. These monkeys participated in our previous study (*Tsunada et al., 2016*), but the behavioral and neuronal data presented here have not been reported previously.

### Idiosyncratic choice-bias behavior

Both monkeys' choice accuracy and RTs depended systematically on stimulus coherence (monkey T: $n = 29$ behavioral sessions; monkey A: $n = 39$ behavioral sessions; *Figure 2a,b*). For high-coherence stimuli, both monkeys almost always reported the correct answer with relatively short RTs. As absolute coherence decreased, performance accuracy decreased and RT increased. These psychometric (choice) and chronometric (RT) data were well described jointly by a drift-diffusion model (DDM; *Figure 2a,b*, pink lines; *Gold and Shadlen, 2007*; *Ratcliff and McKoon, 2008*). The DDM describes the process of forming a decision by accumulating incoming auditory (sensory) evidence over time to one of two pre-defined boundaries and accounts for both the choice (which boundary was reached) and the decision time (when the boundary was reached). These fits provided a consistently good match to the data (average deviance was 11.82 for monkey T [$\chi^2$ -cumulative distribution, $p<0.001$] and 10.99 for monkey A [$p<0.001$; *Wichmann and Hill, 2001*). This result indicates that the monkeys had consistent decision strategies, which were independent of the slight session-by-session differences in the choice of low- and high-frequency stimuli.

Moreover, these fits partitioned the monkeys' RTs into decision and non-decision times (*Figure 2a,b*), which facilitated our ability to identify the contributions of vlPFC activity to decision-making (*Cohen and Newsome, 2009*). In general, decision times for both choices increased as the absolute stimulus coherence decreased, which is a typical feature of the DDM (*Gold and Shadlen, 2007*). In contrast, non-decision times tended to be strongly asymmetric for the two choices, which reflected differences in the speed and preparation time of the right versus left joystick movements produced by the two monkeys. In subsequent analyses, we define the time of the 'decision

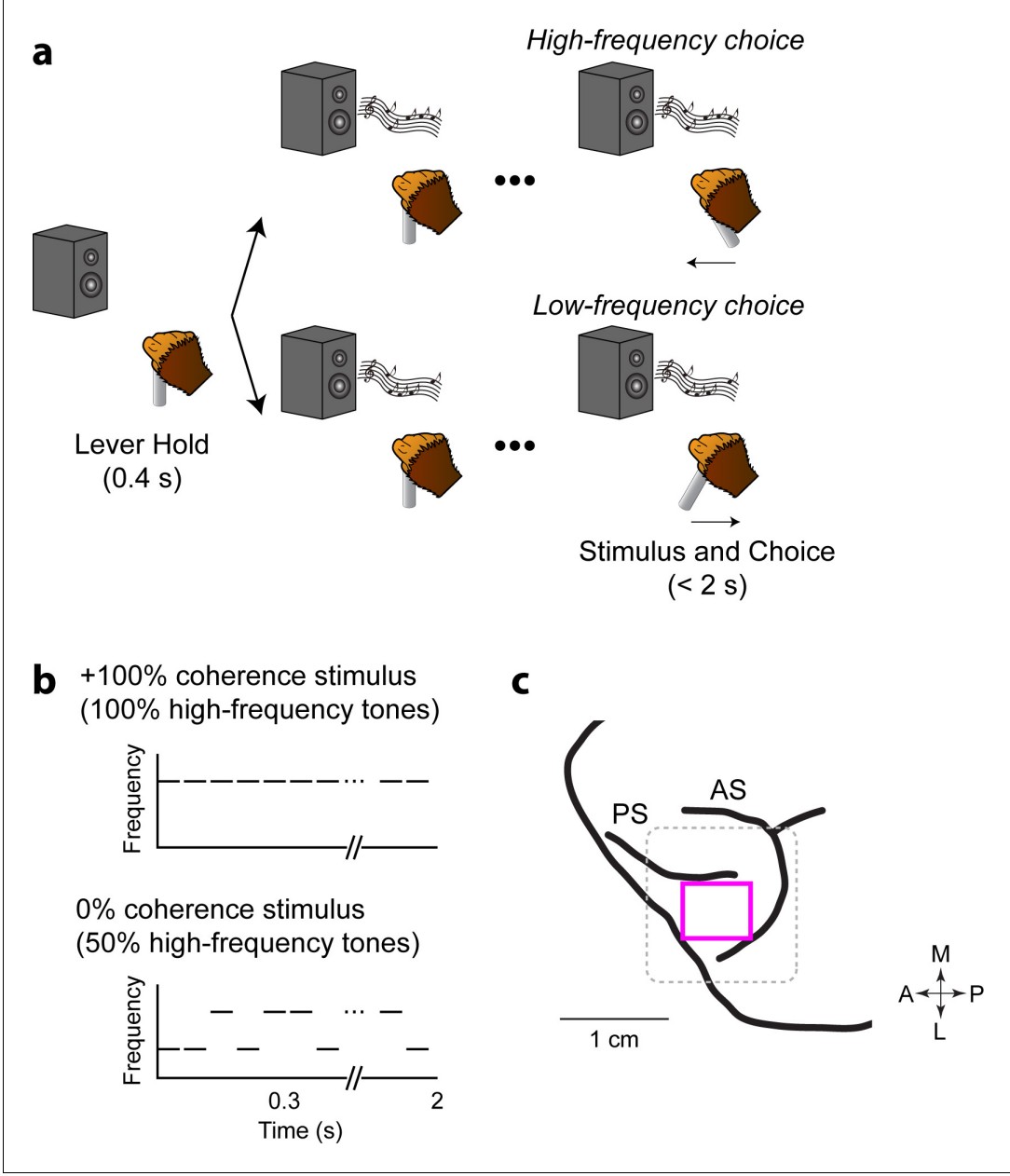

**Figure 1.** Task and stereotactic location of vlPFC. (a) Each monkey decided whether a temporal sequence of tone bursts was predominantly 'low frequency' or 'high frequency' and responded with a rightward or leftward movement, respectively, of the joystick. The monkey could report its choice any time after stimulus onset. (b) Schematics of the auditory stimulus (+100% and 0% coherence stimuli). The auditory stimulus consisted of a sequence of tone bursts (50 ms duration; 10 ms inter-burst interval). Stimulus coherence refers to the percentage of high-frequency bursts (up to +100%) or low-frequency bursts (down to −100%). (c) vlPFC (pink square) is ventral to the posterior aspect of the principal sulcus (PC) and anterior to the arcuate sulcus (AS; *Romanski and Goldman-Rakic, 2002*). The dotted box indicates the circumference of the recording chamber. Arrows indicate the anterior (A)-posterior (P) axis and the medial (M)-lateral (L) axis.
DOI: https://doi.org/10.7554/eLife.46770.002

commitment' as the end of the decision time plus an additional 50 ms to account for stimulus encoding (*Tsunada et al., 2016*).

We also identified idiosyncratic sequential choice biases for the two monkeys. Monkey T tended to use a 'win-stay, lose-switch' strategy (top panels in *Figure 2c,d*). That is, this monkey tended to

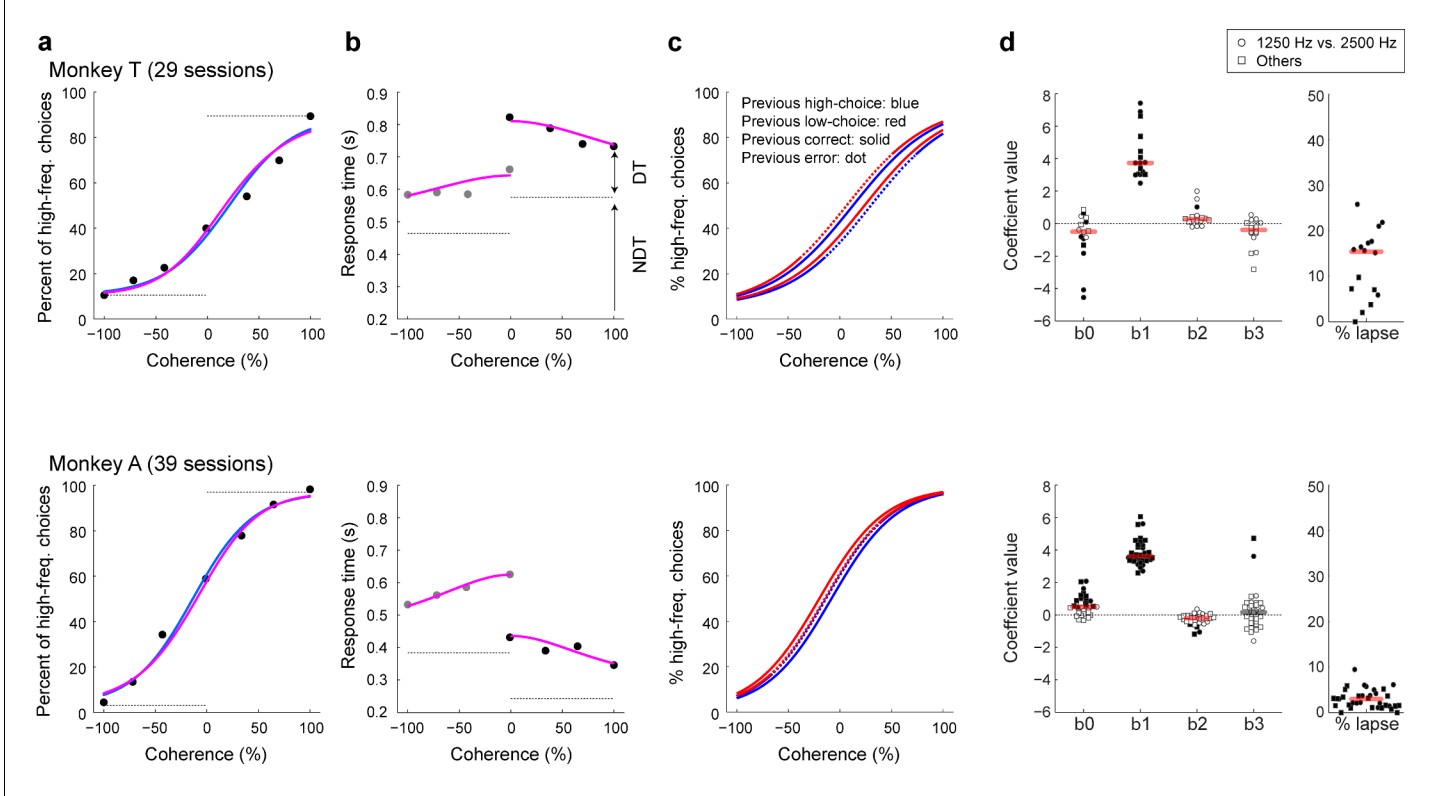

**Figure 2.** Psychophysical performance on the low-high task. Psychometric (**a**) and chronometric (**b**) functions for Monkey T (top) and Monkey A (bottom). These functions were generated from their responses on the current trial. Psychometric functions are plotted as the percentage of trials in which a monkey chose 'high frequency' as a function of signed coherence, in which larger negative/positive coherence values indicate more low/high frequency tone bursts. The horizontal gray lines on the psychometric plots indicate lapse rates (errors for strong stimuli, presumably reflecting lapses in attention or inappropriate application of the decision-motor mapping), which were estimated from logistic fits (solid blue lines). Chronometric functions are plotted using the mean RT, which was the time interval between stimulus onset and onset of joystick movement. Gray dots are low-frequency choices, and black dots are high-frequency choices. Solid pink curves are simultaneous fits of both the psychometric and chronometric data to a drift-diffusion model (DDM). The horizontal dashed gray lines on the chronometric plots indicate choice-dependent non-decision times (NDT) estimated by the DDM fits. Decision times (DT) were estimated as the difference between the trial-specific RT and the choice-specific NDT. (**c**) Psychometric functions computed separately for different sequential conditions, as indicated in the top panel. (**d**) Distributions of best-fitting, session-by-session beta coefficients ($\beta_0$, overall choice bias; $\beta_1$, sensitivity to coherence; $\beta_2$, the tendency to repeat a correct choice; and $\beta_3$, the tendency to repeat an erroneous choice) and lapse rates from the logistic fits. Circles indicate data from sessions using 1250 and 2500 Hz as low and high frequencies, respectively; squares indicate data from other sessions (note that the two conditions corresponded to differences in lapse rates for monkey T but little effect on the other model parameters). Filled data points indicate likelihood-ratio test, $H_0$: regression coefficient equals 0, $p<0.05$. Horizontal bars indicate median values; red bars indicate Wilcoxon sign-rank test, $H_0$: median value equals 0, $p<0.05$.

DOI: https://doi.org/10.7554/eLife.46770.003

repeat the previous choice if that choice was rewarded but switched choices if the previous choice was not rewarded. This tendency was seen as stay-switch biases with positive values following rewarded trials and negative values following non-rewarded trials for both the pooled data (*Figure 2c*; logistic regression: $\beta_0$ [high/low bias]$=-0.46$, $p<0.01$; $\beta_1$ [stimulus coherence]$=2.50$, $p<0.01$; $\beta_2$ [stay/switch bias given that the previous trial was rewarded]$=0.14$, $p<0.01$; $\beta_3$ [stay/switch bias given that the previous trial was not rewarded]$=-0.30$, $p<0.01$) and for the session-by-session data (*Figure 2d*; median $\beta_0 = -0.50$, Wilcoxon sign-rank test, $p=0.04$; $\beta_1 = 3.73$, $p<0.01$; $\beta_2 = 0.27$, $p<0.01$; $\beta_3 = -0.39$, $p=0.03$).

In contrast, monkey A tended to use a 'win-switch' strategy (bottom panels in *Figure 2c,d*). That is, this monkey tended to switch choices following a rewarded choice. Once again, this result was evident in both the pooled data (*Figure 2c*, $\beta_0 = -0.45$, $p<0.01$; $\beta_1 = 3.16$, $p<0.01$; $\beta_2 = -0.19$, $p<0.01$; $\beta_3 = -0.02$, $p=0.88$) and in the session-by-session data (*Figure 2d*, $\beta_0 = 0.50$, $p<0.01$;

$\beta_1 = 3.63$, p<0.01; $\beta_2 = -0.19$, p<0.01; $\beta_3 = 0.17$, p=0.63). For both monkeys, we could not identify similar systematic effects on sequential RTs.

## Post-decision neuronal representations of choice, outcome, and stimulus strength

Identifying the neuronal substrates of a perceptual decision typically involves identifying at least three forms of selectivity: (1) for choice, reflecting the consequence of the decision process; (2) for whether the choice was correct or an error, reflecting a closer association with perception than just sensory or motor processing; and (3) for stimulus strength because the process of forming the decision should reflect not just the categorical choice but also the strength of the evidence used to arrive at that choice (*Gold and Shadlen, 2007*). As detailed below, we identified all three forms of selectivity in vlPFC activity but only after the time of decision commitment on the current trial.

Individual vlPFC neurons had task-driven activity that was modulated selectively by the monkeys' low- versus high-frequency choices (single-unit examples are shown in *Figure 3* and summary population data are shown in *Figure 4*). Across the population of recorded neurons, the onset of choice selectivity for individual neurons (which were recorded in separate sessions and across the two monkeys) spanned the time from the inferred decision commitment through the motor response (joystick movement) and to the time when the reward was delivered or withheld. This selectivity included preferences for both high- and low-frequency choices (corresponding to ipsilateral and contralateral choices, respectively, because high-/low- frequency choices were indicated with leftward/rightward movements, and we recorded from the left hemisphere in both monkeys). We identified 13 and 15 high-frequency (ipsilateral)-preferring neurons in monkeys T and A, respectively; and 10 and 14 low-frequency (contralateral)-preferring neurons, respectively (there was not any evidence for laterality: $\chi^2$-test for $H_0$: no difference in the proportion of contralateral- and ipsilateral preferring neurons, p>0.05 for both monkeys).

The choice selectivity of individual neurons was also affected by the outcome of the current trial. Specifically, choice selectivity tended to be higher on correct trials than on error trials (*Figure 5*). This difference in choice selectivity was apparent even before reward delivery and thus could not be explained trivially as a direct response to reward delivery. Instead, this difference likely reflected differences in decision processing on correct versus error trials, including more uncertainty in the neuronal representation of the sensory evidence and therefore possibly lower reward expectations on error trials (*Gold and Shadlen, 2003*).

In contrast to these single-neuron modulations by choice and outcome, the responses of individual vlPFC neurons were not selective for stimulus coherence. In particular, we found that, at any given time point, at most six vlPFC neurons were modulated by stimulus coherence for either preferred or non-preferred choices, which is not more than would be expected by chance in our sample (*Figure 4b*). However, we found a more robust representation of stimulus coherence at the level of population neuronal activity. The intuition for this discrepancy can be seen in *Figure 4a*: the fraction of neurons with choice selectivity was systematically smaller with decreasing coherence, implying that population-level signals were dependent on coherence. Although these effects did not correspond to statistically reliable differences in single-neuron coherence selectivity, there was enough information across the population of 103 neurons recorded from all sessions from both monkeys for a linear classifier to decode both stimulus coherence and choice (*Figure 6*). In both cases, decoding accuracy rose above chance levels only after the end of the decision process (blue vertical stripes in the stimulus-aligned plots in the left panels of *Figure 6*) and peaked around the time reward delivery, implying a form of post-decision processing.

Thus, the vlPFC population had access to the key features of the decision process, including the strength of the evidence used to form the decision, the choice, and whether the choice was correct or an error. These signals were not apparent before the decision commitment, implying that they did not contribute to the formation of the current decision. Instead, the timing of these signals suggests that they may play a role in post-decision processing that can link one decision to the next decision.

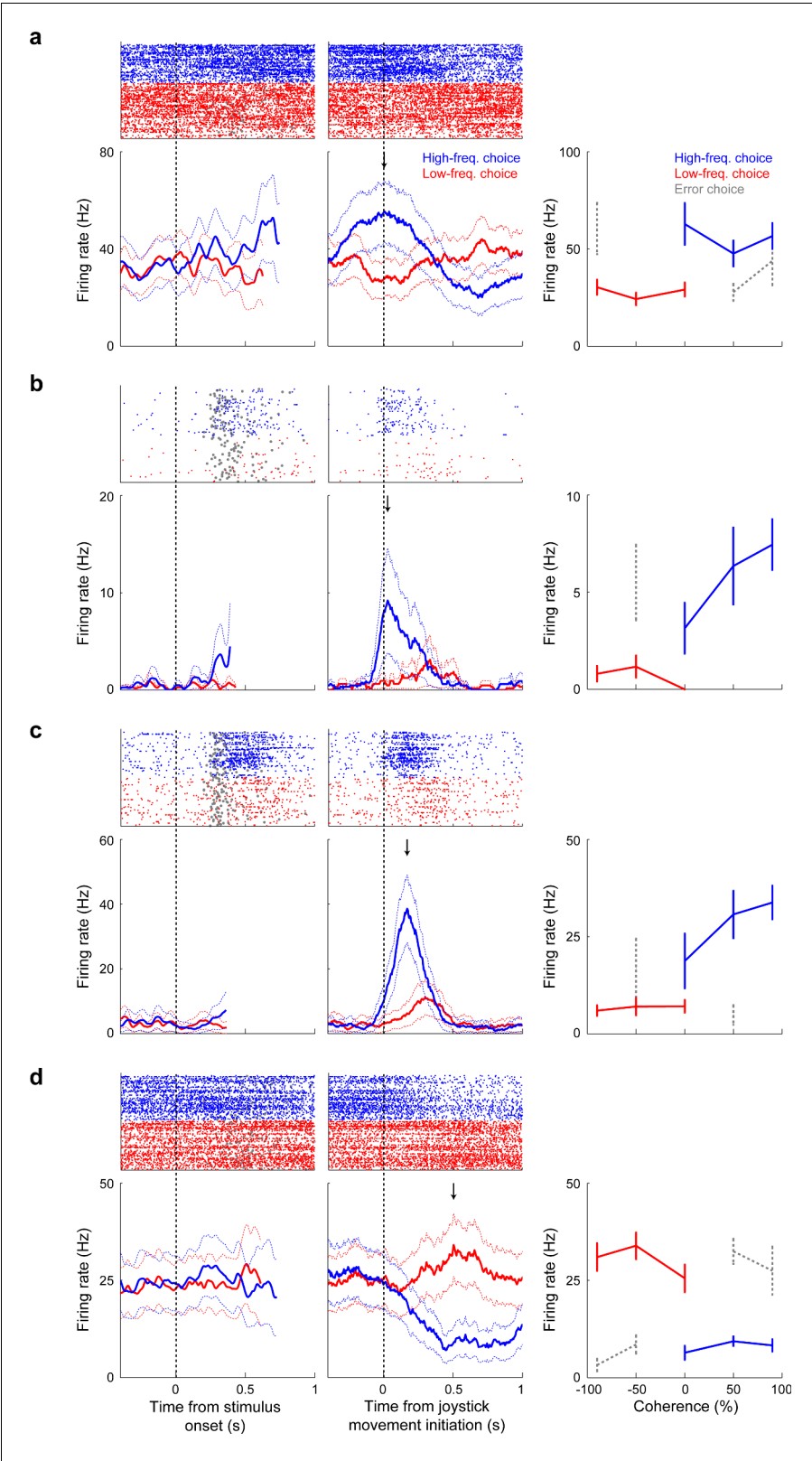

**Figure 3.** Neuronal sensitivity to choice in single vlPFC neurons. (**a–d**) The left plots are raster and peristimulus-time histograms from correct trials only showing sensitivity to low-frequency (<0% coherence; red) and high-frequency choices (>0% coherence; blue). The thick lines indicate mean firing rate, and the dotted lines indicate the 95% confidence intervals. Data are aligned relative to stimulus onset. Gray circles in the raster plots indicate the time of onset of joystick movement. The middle plots show the responses of the same neurons but aligned relative to the onset of joystick movement. The

*Figure 3 continued on next page*

*Figure 3 continued*

arrow indicates the time of peak choice selectivity. The right plot summarizes each neuron's firing rate during its peak firing rate ±100 ms: correct low-frequency choices are shown in red, high-frequency choices in blue, and incorrect choices in gray (only for coherences with at least five trials). Error bars indicate the standard error of the mean.

DOI: https://doi.org/10.7554/eLife.46770.004

## Post-decision vlPFC activity encodes subsequent choice biases

Neuronal activity in the post-decision epoch was selective for both the current choice and the subsequent choice. This selectivity for the subsequent choice also matched each monkey's idiosyncratic choice biases (*Figure 2c,d*), particularly following rewarded trials. Specifically, monkey T's tendency to repeat rewarded trials was reflected in post-decision neuronal responses that tended to be slightly larger on trials in which the subsequent choice matched the neuron's choice selectivity (*Figure 7* top). For example, if a neuron tended to respond more for a high-frequency choice in the post-decision epoch of the current trial, its response tended, on average, to be slightly higher when the monkey made a high-frequency choice on the subsequent trial. In contrast, monkey A's tendency to switch after rewarded trials was reflected in neuronal responses that tended, on average, to be slightly smaller on trials in which the subsequent choice matched the neuron's choice selectivity (*Figure 7* bottom). These effects had slightly different time courses in the two monkeys. Nonetheless, in both cases, these effects occurred after the decision commitment on correct trials, which corresponds to a time period when evaluative processing could be used to adjust subsequent decisions. We could not identify similarly reliable effects following error trials, possibly reflecting the much smaller data sets from those trials.

## Electrical microstimulation biased the subsequent choice

We used electrical microstimulation to test whether vlPFC activity plays a causal role in driving choice biases on the subsequent trial. We applied microstimulation from the time of stimulus onset until just after the behavioral response on a randomly selected 50% of trials in a subset of sessions (*n* = 11 sessions for monkey T, 21 sessions for monkey A). Despite the fact that this protocol was designed to test our initial hypothesis that vlPFC activity encoded formation of the current decision (and thus microstimulation was applied primarily during decision formation), we found that microstimulation did not systematically affect either the choice bias or the sensitivity of the decision on the current trial (single-site examples are shown in *Figure 8a,b*; population summaries are shown in *Figure 8c,d*).

Instead, we found that microstimulation induced choice biases for the subsequent decision that depended systematically on the choice selectivity of the recorded vlPFC neuron at the microstimulation site. If microstimulation was applied at a site that was tuned for low-frequency choices, it tended to cause a low-choice bias on the subsequent trial (single-site example in *Figure 8e*). In contrast, if microstimulation was applied at a site that was tuned for high-frequency choices, it tended to cause a high-choice bias on the subsequent trial (single-site example in *Figure 8f*). Accordingly, across the population of microstimulation sites from both monkeys, the induced choice bias on the subsequent trial was correlated positively with the strength and direction of choice selectivity at the given site (*Figure 8g*). In other words, microstimulation at a site with neuronal activity that was selective for a low- or high-frequency choice on the current trial biased the monkeys' choices toward a low- or high-frequency choice, respectively, on the subsequent trial. Further, the absolute magnitude of this bias was positively correlated with the strength of choice selectivity at the site of microstimulation. We did not identify any concomitant, systematic changes in psychometric sensitivity (*Figure 8h*).

These microstimulation effects also depended on specific choice patterns, albeit slightly differently for the two monkeys. When microstimulation was applied on a trial that resulted in a rewarded high choice, the subsequent choice tended to be biased in the same direction as the choice selectivity of the neuron recorded at the site of microstimulation for both monkeys (monkey T: Spearman's rank correlation coefficient ρ = 0.90, p<0.001; monkey A: ρ = 0.61, p=0.04). When microstimulation was applied on a trial that resulted in a rewarded low choice, a similar effect was found only for one of the two monkeys (monkey T: ρ = 0.91, p<0.001; monkey A: ρ = 0.10, p=0.75). Together, these

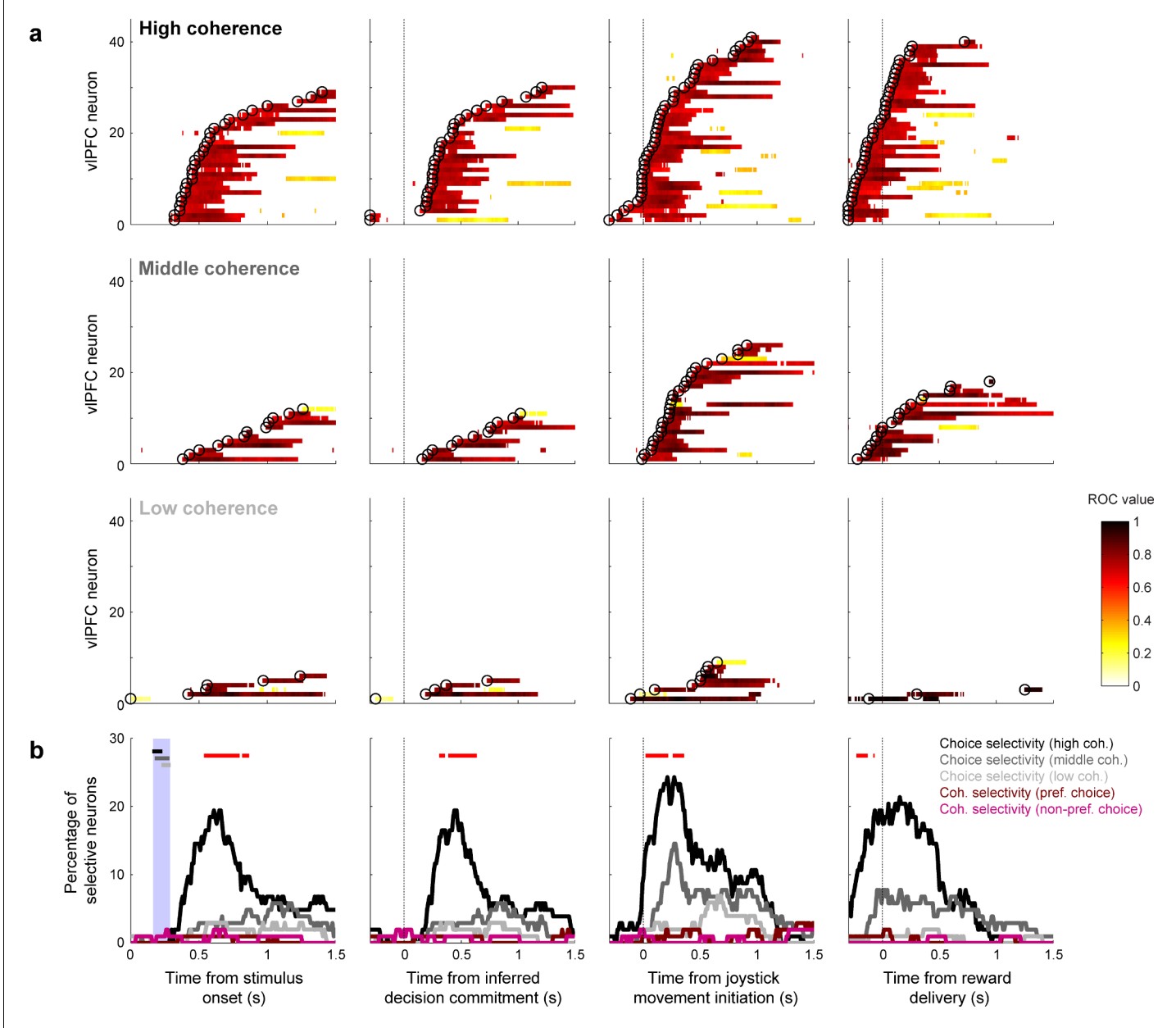

**Figure 4.** Population selectivity for vlPFC neurons. (a) Summary of choice selectivity. Data from individual neurons are sorted by the onset of choice selectivity (open circles), defined as the first of three consecutive time bins with reliably different responses for the two choices (Wilcoxon rank-sum test, $H_0$: no median difference in firing rates for the two choices, p<0.05, FDR corrected). Color indicates the ROC value of choice selectivity from correct trials (see legend). Rows show data for high (<−80% versus >+80%), middle (−80% to −20% versus +80% to+20%), and low (−20% to 0 versus 0 to +20%) coherence trials, as indicated. (b) Percentage of neurons with significant selectivity for choice or coherence (Wilcoxon rank-sum test for $H_0$: no median difference in firing rates elicited by high- versus middle- coherence stimuli for each choice, p<0.05, FDR corrected) computed in 300 ms time bins with 10 ms steps. Choice selectivity is shown separately for high, middle, and low coherences, as indicated. Red points indicate times corresponding to a significant difference in the proportion of choice-selective neurons at each coherence level (running $\chi^2$-test for $H_0$: proportion is the same, p<0.05, FDR corrected). Coherence selectivity is shown in dark red for preferred choices (i.e., the choice direction that elicits higher firing rates) and light red for non-preferred choices. In the leftmost panel, the horizontal bars represent the range of the inferred times of the decision commitment for high (black), middle (dark gray), and low (light gray) coherence stimuli (the range is indicated by the large vertical bar). In (a) and (b), the data in each panel are aligned relative to different task epochs (from left to right): stimulus onset, inferred decision commitment, onset of joystick movement, and time of reward delivery.

DOI: https://doi.org/10.7554/eLife.46770.005

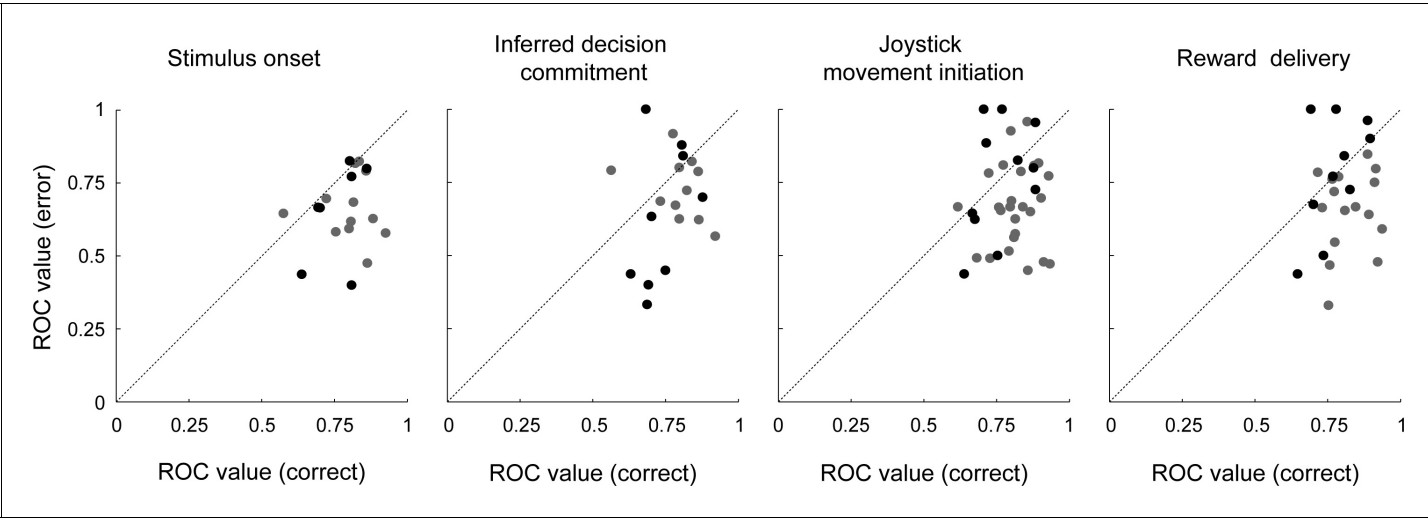

**Figure 5.** Choice selectivity on correct and error trials. Scatterplots showing, on a neuron-by-neuron basis, the peak ROC choice-selectivity value computed on correct versus error trials. Both values were computed from spiking data occurring at the time of peak ROC-based choice selectivity from correct trials for the given neuron. Black/gray points correspond to data from high/middle coherence stimuli. The line in each panel is the line of unity. The panels show data computed relative to different task epochs (from left to right): stimulus onset, inferred decision commitment, onset of joystick movement, and time of reward delivery. Across all epochs, error ROC values tended to be smaller than correct ROC values (Wilcoxon sign-rank test for $H_0$: median ROC values are the same, $p < 0.05$). Different panels have different numbers of data points because for some sessions, there were not enough trials to reliably calculate the error ROC.

DOI: https://doi.org/10.7554/eLife.46770.006

effects are consistent with the hypothesis that vlPFC activity is causally involved in evaluative processing that adjusts subsequent decisions.

## Discussion

We combined behavioral modeling, neuronal recordings, and electrical microstimulation to identify causal contributions of the primate vlPFC to a simple auditory perceptual decision about the frequency content of a sequence of tone bursts. vlPFC population activity had many of the hallmarks of a decision variable that could account for the monkeys' patterns of choices and RTs, including selectivity for the strength of the sensory evidence, the monkey's choice, and whether the choice was correct or incorrect. However, these forms of selectivity were not evident until after the time of the decision commitment and thus were inconsistent with a role for these neurons in forming the current decision. Instead, this post-decision selectivity appeared to support sequential adjustments to the decision process. Specifically, post-decision neuronal activity was modulated by each monkey's idiosyncratic sequential choice biases. Further, electrical microstimulation at vlPFC sites affected the monkeys' choices on the subsequent, but not the current, trial. Together, these results imply a role for post-decision vlPFC activity in encoding information about the process of forming the just-completed decision that is used to generate individualized biases that affect the subsequent decision.

These post-decision signals are a form of 'decision-trace' activity that has been identified in numerous brain areas, including parts of the prefrontal and parietal cortices (*Barraclough et al., 2004*; *Bizzi, 1968*; *Funahashi et al., 1991*; *Tsujimoto and Sawaguchi, 2004*; *Ding and Gold, 2012a*; *Hwang et al., 2017*; *Akrami et al., 2018*; *Ding and Gold, 2010*; *Histed et al., 2009*). This kind of activity represents information about the immediately preceding decision that can be used as part of a feedback-driven learning process to adjust future decisions based on a comparison between the expected and actual outcome of prior decisions (*Sutton and Barto, 1998*). Consistent with this idea, the population-level post-decision representations that we identified in the vlPFC are, in principle, sufficient to compute a confidence or reward-expectation signal for the just-completed decision (*Ding and Gold, 2010*; *Ding and Gold, 2012a*; *Kiani and Shadlen, 2009*; *Kepecs et al., 2008*). This signal could then be compared to the actual outcome to adjust the subsequent decision,

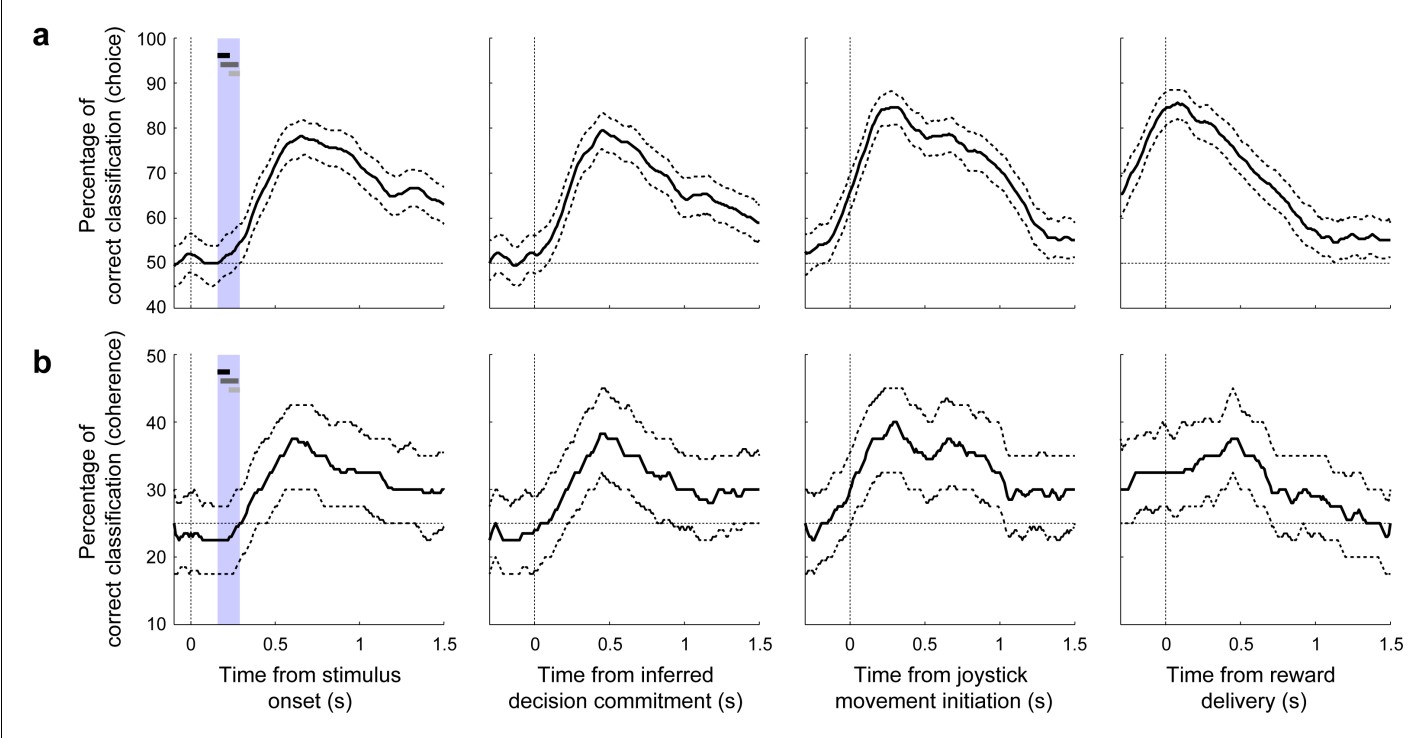

**Figure 6.** Classifier analysis. The ability of a linear classifier to determine from the population of vlPFC neurons the: (a) current choice (low frequency [<0%] or high frequency [>0%]), or (b) stimulus coherence in four bins (<−50%, −50–0%, 0%–+50%, or >+50%). Results were computed using correct trials only in 300 ms time bins with 10 ms steps. Thick lines represent median decoding performance; dashed lines are the interquartile range. In the leftmost panel, the horizontal bar represents the range of the inferred times of the decision commitment for high (black), middle (dark gray), and low (light gray) coherence (the range is indicated by the large vertical bars). Choice- and coherence-decoding performance is aligned relative to different task epochs (from left to right): stimulus onset, inferred decision commitment, onset of joystick movement, and time of reward delivery. We did not conduct a classifier analysis on error trials because there was not enough data to generate reliable results.

DOI: https://doi.org/10.7554/eLife.46770.007

possibly in other cortical or subcortical brain regions known to encode reward feedback and prediction error (*Schultz, 2015*).

The fact that our two monkeys used different biasing strategies (win-stay for monkey T, win-switch for monkey A) affects the interpretation of our findings in two primary ways. First, vlPFC selectivity for the subsequent choice was consistent with each monkey's strategy. This result provides stronger support for the behavioral relevance of these signals than if they encoded features of behavior that were only present on average. Second, our microstimulation effects on the subsequent choice tended to depend on the choice tuning of neurons at the site of microstimulation and not the idiosyncratic sequential bias strategy of the monkey. This result implies that the vlPFC provides a choice-dependent signal that is used to generate sequential biases but may not participate directly in forming the idiosyncratic strategies that use those biases.

These findings are broadly consistent with recent studies that have shown a role for the vlPFC in strategy-switching and probabilistic-learning tasks, which share features of the sequential effects that we identified (*Baxter et al., 2009*; *Rudebeck et al., 2017*). However, our findings of primarily post-decision processing in the vlPFC are somewhat inconsistent with other studies that have implicated the PFC in forming auditory and other decisions (*Russ et al., 2008b*; *Cohen et al., 2009*; *Lee et al., 2009*; *Bizley and Cohen, 2013*; *Kim and Shadlen, 1999*). The reasons for this discrepancy are not clear. One possible reason is that we sampled a different PFC population than in these other studies, and these different populations make different contributions to decision versus post-decision processing. Another possibility is that previous reports of decision-related activity were also largely post-decisional. However, accounting for this possibility is not straightforward: the previous studies did not use RT tasks, making it difficult to interpret those results in terms of whether the

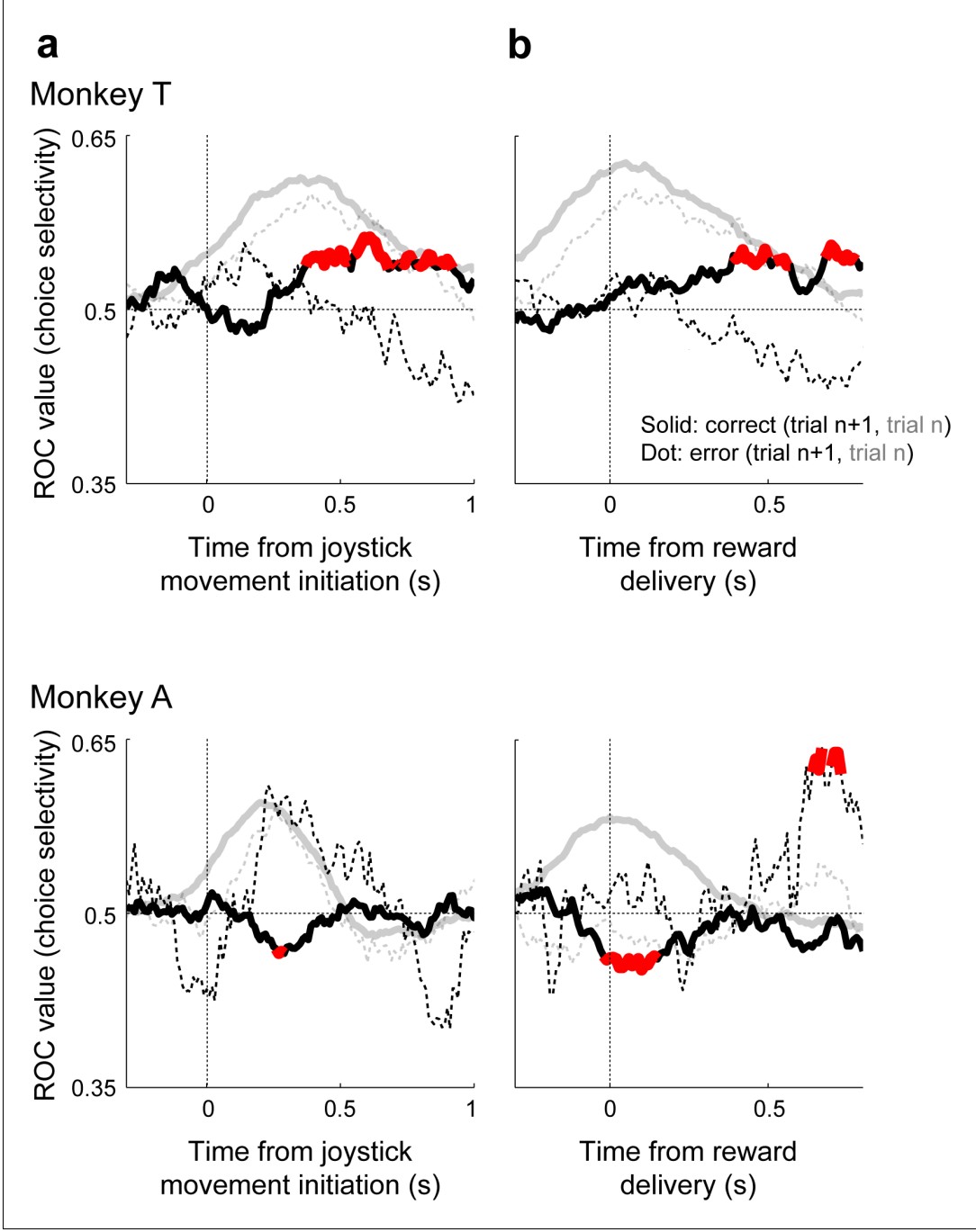

**Figure 7.** Choice selectivity for the current and next trial. For Monkey T (top) and Monkey A (bottom), choice selectivity is plotted as a function of time relative to the onset of joystick movement (**a**) and reward delivery (**b**). Lines indicate ROC-based choice selectivity computed in 300 ms time bins, with 10 ms steps from pooled spiking data across all recorded neurons (z-scored per neuron) that contributed at least 121 for monkey T and 54 trials for monkey A under the given conditions. Solid/dotted lines correspond to correct/error outcomes on the current trial. Black lines indicate selectivity for repeated (ROC values > 0.5) versus switched (<0.5) choices on the next trial, relative to the choice on the current trial (i.e., values > 0.5 imply that the neuronal population tended to respond more in anticipation of a repeated choice). For reference, gray lines indicate selectivity for the preferred choices on the current trial (i.e., values > 0.5 indicate, by definition, selectivity for the choice that elicited the larger average spike rate during peak firing rate ±100 ms for each neuron). Red points, computed only for the black curves, indicate permutation test for $H_0$: ROC value equals 0.5, p<0.05.
DOI: https://doi.org/10.7554/eLife.46770.008

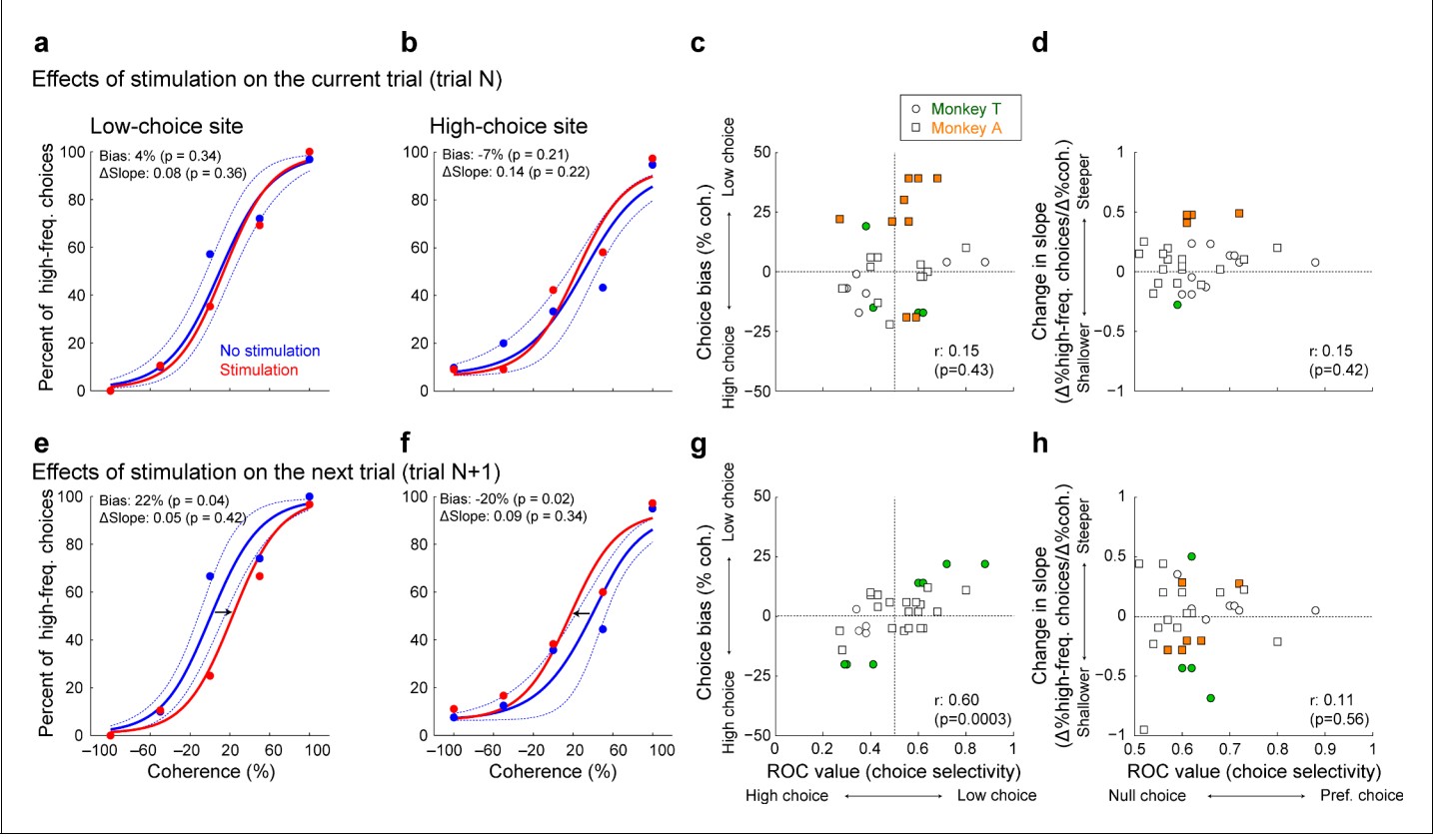

**Figure 8.** Effect of microstimulation on behavioral performance on the current and next trial. (**a** and **b**) Single-site examples of the effects of vlPFC microstimulation on psychometric performance on the current trial for a low-choice site (**a**) and a high-choice site (**b**). Psychometric functions are plotted as in *Figure 2*. Red/blue symbols are for data from trials with/without microstimulation. Solid lines are logistic fits, computed separately for the two conditions. Dotted lines are 95% confidence intervals of the non-microstimulation trials that were calculated by a bootstrap procedure (*Ding and Gold, 2012b*). (**c** and **d**), Scatterplots showing session-by-session effects of microstimulation on the correlation between neuronal choice selectivity and the percent change in psychometric choice bias (**c**); Spearman's rank correlation coefficient ρ = 0.15, p=0.43) and the change in psychometric slope (**d**); ρ = 0.15, p=0.42) of the current decision. (**e** and **f**) Single-site examples of microstimulation's effects on psychometric performance on the next trial for a low-choice site (**e**) and a high-choice site (**f**). The data are formatted in the same manner as panels (**a**) and (**b**). (**g** and **h**), Scatterplots show session-by-session effects of microstimulation on the correlation between neuronal choice selectivity and the percent change in psychometric choice bias (**g**; ρ = 0.60, p=0.0003) and the change in psychometric slope (**h**; ρ = 0.11, p=0.56) of the next decision. Filled data points are significant single-session microstimulation-induced changes in the given psychometric property (permutation test, p<0.05).

DOI: https://doi.org/10.7554/eLife.46770.009

reported decision-related signals occurred before or after the decision was formed on each trial (*Russ et al., 2008b*; *Cohen et al., 2009*; *Fritz et al., 2010*).

We still do not know the brain regions that form the decision for our task and consequently do not understand the mechanisms underlying these decisions. Because the monkeys' choice and RT patterns reflected both the temporal accumulation of sensory evidence and sequential choice biases, we would expect these putative brain regions to implement two key operations. First, they should temporally accumulate the sensory evidence that is represented in the auditory cortex, particularly AL (*Tsunada et al., 2016*). Second, they should combine this accumulated evidence with historical sensory, outcome, and choice information, similar to that represented in vlPFC, to drive sequential choice biases (*Ding and Gold, 2012a*; *Ding and Gold, 2010*; *Kim and Shadlen, 1999*; *Hwang et al., 2017*; *Akrami et al., 2018*). One possibility is a set of other brain areas that have shown information-accumulation activity on other tasks, such as the dorsolateral PFC and parts of the posterior parietal cortex (*Gold and Shadlen, 2007*; *Brody and Hanks, 2016*). The posterior parietal cortex is a particularly compelling target of future studies because in rodents, it contributes to history-dependent choice biases on an auditory decision task (*Akrami et al., 2018*). Another

intriguing possibility is an auditory-specific circuit involving the parabelt region of auditory cortex, which receives direct input from AL, projects to the vlPFC, and analyzes acoustic properties of behaviorally relevant sounds (*Hackett et al., 1999*; *Romanski et al., 1999*; *Petkov et al., 2004*).

Future studies that aim to identify neuronal activity related to decision formation for this kind of auditory task will likely benefit from not only an RT design to better identify the temporal epoch of decision formation but also a more thorough understanding of the dynamic and possibly idiosyncratic nature of the computations used to convert incoming sensory evidence into the categorical choice (*Cohen and Newsome, 2009*; *Fan et al., 2018*). These studies might also benefit from analyses that focus on substrates of subject-specific decision strategies, which help to establish the behavioral relevance of the neuronal signals (*Fan et al., 2018*; *Busse et al., 2011*; *Abrahamyan et al., 2016*). For example, in the present study, we found that our monkeys had different strategies of sequential choice biases (win-stay for monkey T versus win-switch for monkey A; *Figure 2*). These kinds of subject-specific choice biases have been reported previously in humans and animals, but their neuronal correlates have yet to be fully explained (*Busse et al., 2011*; *Abrahamyan et al., 2016*; *Ding and Gold, 2010*). Because the monkeys' idiosyncratic behavioral strategies corresponded to different patterns of vlPFC choice selectivity, it implies that these signals may play a behaviorally relevant, subject-specific role in the evaluation and adjustment of the decision process, rather than providing a simple memory trace of common components of the decision process (*Tsujimoto and Postle, 2012*; *Tsujimoto and Sawaguchi, 2004*).

It is worth emphasizing that in our study, the representations in the vlPFC of critical decision-related variables (i.e., choice, outcome, and the strength of the sensory evidence) were not all evident in the spiking activity of individual neurons but instead were seen at the level of neuronal populations. The time course of these representations was also distributed across the neuronal population: different neurons (in our case, recorded in separate sessions) responded selectively at relatively restricted times with onsets that tiled the task epoch in a manner similar to other reports of working memory in the prefrontal cortex (*Zaksas and Pasternak, 2006*; *Jun et al., 2010*; *Lundqvist et al., 2016*; *Brody et al., 2003*; *Schmitt et al., 2017*). These population-level representations highlight the importance of conducting population recordings and analyses to identify and understand complex decision-related computations in the brain (*Murray et al., 2017*; *Pouget et al., 2000*; *Kohn et al., 2016*; *Averbeck et al., 2006*; *Meyers, 2018*; *Meister et al., 2013*).

In addition to recording from large neuronal populations, it would be instructive to examine vlPFC activity under a broader range of task conditions to better understand its general role in the sequential processing of auditory information. Our findings suggest that vlPFC can support auditory processing across trials. Other studies have shown that vlPFC also can play a role in rule-based sequence learning, which requires complex temporal processing over multiple time scales (*Wilson et al., 2015*). This kind of flexible temporal processing has been associated with human syntactic processing, a key feature of language (*Wilson et al., 2015*; *Kikuchi et al., 2017*; *Wilson et al., 2017*). It would be interesting to explore whether monkey vlPFC may possess a precursor system used in human language.

## Materials and methods

The University of Pennsylvania Institutional Animal Care and Use Committee approved all of the experimental protocols, which were conducted under protocol 804699. All surgical procedures were conducted using aseptic surgical techniques and with the monkeys kept under general anesthesia. A transparent reporting form is available. The authors were not blind to group allocation during the experiment and when assessing the data outcomes.

Two male monkeys (*Macaca mulatta*; monkey T [15 years old] and monkey A [14 years old]) participated in this study. Both were used in a previous study of auditory cortex (*Tsunada et al., 2016*), and monkey T was also used in a previous study of vlPFC (*Tsunada et al., 2011a*). In each session, the monkey was seated in a primate chair. A calibrated speaker (model MSP7, Yamaha) was placed in front of the monkey at eye level. The monkey moved a joystick, which was attached to the primate chair, to indicate their behavioral report. All experimental sessions took place in an RF-shielded room that had sound-attenuating walls and echo-absorbing foam on the inner walls.

## Identification of ventrolateral prefrontal cortex

Prior to implantation of a recording chamber, the stereotactic location of vlPFC, which includes Brodmann area 45 and 46 (*Figure 1c*), was identified through structural MRI scans (*Frey et al., 2004*; *Johnston et al., 2016*). We centered the recording chamber over this cortical location on the left hemisphere for both monkeys. vlPFC was further identified by its auditory responses (*Romanski and Goldman-Rakic, 2002*; *Russ et al., 2008a*).

## Auditory tasks and stimuli

Auditory stimuli were generated using Matlab (The Mathworks Inc) and the RX6 digital-signal-processing platform (TDT Inc).

### Frequency tuning

We measured the frequency tuning of vlPFC recording sites by presenting individual tone bursts in a random order while the monkey listened passively. The tone bursts (100 ms duration with a 5 ms $\cos^2$ ramp; 65 dB SPL) varied between 0.3–12 kHz in one-third octave steps. The monkeys did not receive any rewards during this time period.

### Low-high task

The low-high task was a single-interval, two-alternative forced-choice discrimination task that required a monkey to report whether a temporal sequence of tone bursts contained more low-frequency or high-frequency tone bursts (*Figure 1a,b*). A trial began with the monkey grasping the joystick. After a 400 ms delay, we presented a sequence of tone bursts (50 ms duration; 5 ms $\cos^2$ ramp; 10 ms inter-burst interval). The monkey moved the joystick: (1) to the right to report that the sequence contained more low-frequency tone bursts, or (2) to the left to report that the sequence contained more high-frequency tone bursts. The monkey could report its choice at any time after stimulus onset.

On a trial-by-trial basis, we randomly varied the proportion of low- and high-frequency tone bursts (coherence) in the auditory stimulus. We varied coherence from −100% (all low-frequency tone bursts) to +100% (all high-frequency tone bursts), with 0% coherence corresponding to 50% of the tone bursts randomly assigned as low or high frequency. Based on each trial's coherence, a tone-burst sequence was generated by randomly assigning the frequency of each tone burst to the low- or high-frequency value.

All correct choices were rewarded with a drop of juice. For trials with ambiguous stimuli (between −20% and +20% coherence), the monkey was rewarded on 50% of randomly selected trials, independent of their behavioral report. The monkey's reward did not depend on the speed of their behavioral report, only its accuracy. Errors resulted in an increased (by 2 s) inter-trial interval.

During testing, we generally used 1250 and 2500 Hz as the low and high frequencies, respectively ($n$ = 12 out of 29 sessions for monkey T, 17 out of 39 sessions for monkey A). Otherwise, we used other values, with low/high values always less/greater than 1750 Hz, and with the two values in a given session always separated by 1–3 octaves.

## Recording methodology

At the start of each recording session, a tungsten microelectrode (~1.0 MΩ @ 1 kHz; FHC Inc) or a tetrode (0.5–0.8 MΩ @ 1 kHz; Thomas RECORDING GmbH) was placed in a skull-mounted microdrive (Narishige, MO-95) and then lowered into the brain through a recording chamber. All neuronal signals were sampled at 24 kHz, band-pass filtered between 0.7–7.0 kHz (RA16PA and RZ2, TDT Inc), and stored for online and offline analyses. OpenEx (TDT Inc), Labview (NI Inc), and Matlab (The Mathworks) software synchronized behavioral control with stimulus production and data collection. Single-neuron activity was isolated from the neuronal signals with on-line (OpenSorter, TDT Inc) and off-line (Offline Sorter, Plexon Inc) spike-sorting programs.

## Data-collection strategy

In our initial sessions, once multi-unit spiking activity was detected, we presented tone bursts to generate a frequency-tuning curve. However, because most vlPFC neurons were not frequency tuned (only 3 out of 65 tested sites, Kruskal-Wallis test, $p < 0.05$), we generally used one of three

standardized sets of fixed low and high frequencies: (1) 1000 and 3000 Hz ($n$ = 52 neurons); (2) 1250 and 2500 Hz ($n$ = 40 neurons); and (3) an arbitrary value <1750 Hz and a value 1–3 octaves above the selected low frequency ($n$ = 11 neurons). Next, the monkey participated in the low-high task. We varied stimulus coherence randomly on a trial-by-trial basis.

During sessions with electrical microstimulation, we delivered negative-leading bipolar current pulses (frequency of stimulation: 300 Hz; inter-bipolar-pulse interval ~3 ms; pulse duration: 250 μs; amplitude: 25–75 μA) on 50% of randomly interleaved trials using a dual-output square-pulse stimulator (Grass S88) and two optical isolation units (Grass PSIU6; *Ding and Gold, 2012b*; *Hanks et al., 2006*). Microstimulation started with stimulus onset and terminated at joystick movement. Because microstimulation trials were rewarded using the same schedule as non-microstimulation trials, the monkeys were not incentivized to respond differently during microstimulation trials than during non-microstimulation trials.

## Behavioral analyses

For all analyses, stimulus coherence was calculated from the actual proportion of low- and high-frequency tone bursts that were presented from stimulus onset until the monkey indicated its choice by moving the joystick on a given trial.

### Drift-diffusion model

Psychophysical and chronometric data were fit to a standard drift-diffusion model (DDM), which models a decision process in which noisy evidence is accumulated over time until it reaches a fixed bound (*Brunton et al., 2013*; *Ding and Gold, 2012a*; *Eckhoff et al., 2008*; *Gold and Shadlen, 2007*; *Green et al., 2010*; *Mulder et al., 2013*; *Ratcliff et al., 1999*; *Shadlen et al., 2006*). This version of the model had five free parameters: k, A, B, $F_1$, and $F_2$. k governed the stimulus sensitivity of the moment-by-moment sensory evidence. The evidence had a Gaussian distribution $N$ (μ,1) in which the mean μ scaled with the stimulus coherence (COH): $\mu = k \times COH$. The decision variable was the temporal accumulation of this momentary sensory evidence. A decision occurred when this decision variable reached a decision bound (+A or -B, which corresponded to a high- and low-frequency choice, respectively). 'Decision time' was the time between stimulus onset and the crossing of either bound. Response time (RT; which was the time from stimulus onset to the onset of joystick movement) could also be defined as the sum of this decision time and a 'non-decision time' ($F_1$ for a high-frequency choice and $F_2$ for a low-frequency choice). Non-decision time includes processes such as stimulus encoding and motor preparation. We defined the time of 'decision commitment' as the end of the decision time plus an additional 50 ms to account for sensory latency (*Tsunada et al., 2016*). The probability that the decision variable crossed the +A bound first is $\frac{e^{2\mu B}-1}{e^{2\mu B}-e^{-2\mu A}}$. The average decision time is $\frac{A+B}{\mu} \times \coth(\mu(A+B)) - \frac{B}{\mu}\coth(\mu B)$ for high-frequency choices and $\frac{A+B}{\mu} \times \coth(\mu(A+B)) - \frac{A}{\mu}\coth(\mu A)$ for low-frequency choices.

### Logistic analysis of psychophysical data

We also used a logistic function to fit psychophysical choice data (*Ding and Gold, 2012b*; *Salzman et al., 1990*; *Cox, 1970*). This function related the probability (p) that the monkey reported high-frequency choices as a function of coherence (COH): $p = L + (1-2L)\frac{1}{1+e^{-(\beta_{COH}*COH+\beta_0)}}$. L represents the upper and lower asymptotes (lapse rates) of the logistic function. $\beta_{COH}$ quantifies the effect of coherence on the monkey's choices and governs the slope of the psychometric function. $\beta_0$ quantifies choice biases and governs the function's horizontal position. In a separate analysis, we used indicator variables to determine additional choice biases that were conditioned on the outcome of the previous trial: (1) if the choice on the previous trial was rewarded (rewarded high choice =+1, rewarded low choice = −1, not rewarded = 0), and (2) if the choice on the previous trial was not rewarded (not rewarded high choice =+1, not rewarded low choice = −1, rewarded = 0). If a monkey repeated the same choice, the coefficient values of the indicator variables would be positive. If a monkey switched its choice, the coefficient values would be negative. For our session-by-session analyses (*Figure 2d*), we removed some sessions from this analysis due to the small number of trials per condition, resulting in 16 sessions for monkey T and 35 sessions for monkey A. A maximum-likelihood procedure fit the logistic function to the behavioral data.

To quantify the effects of microstimulation on behavior, we fit the logistic function (with additional indicator variables and assumed that each session had a single lapse rate across all microstimulation and non-microstimulation trials) to choice data from subsets of trials in individual sessions and tested whether the choice bias and perceptual sensitivity differed: (1) when microstimulation was applied on the current trial (+1) versus when it was not applied (0), and (2) when microstimulation was applied on the previous trial (+1) versus when it was not applied (0). 'Choice bias' was defined as the horizontal shift of psychometric functions. More specifically, the shift was calculated as the difference between stimulus coherences that elicited 50% high-frequency choices. 'Perceptual sensitivity' was defined as the change in the slope of the psychometric function determined from the 25% and 75% high-frequency choice points.

## Neuronal analyses

We did not use statistical methods to predetermine sample sizes. Our sample sizes were similar to those reported in previous publications, including our recent study of auditory cortex (*Roitman and Shadlen, 2002*; *Selezneva et al., 2006*; *Tsunada et al., 2016*).

### Single-neuron choice selectivity

To identify if and when each neuron had statistically significant choice-related activity, we performed a running Wilcoxon rank-sum test for each pair of stimulus coherence bins with the same magnitude but different signs ($H_0$: firing rates elicited by the coherence pair are the same, $p < 0.05$, FDR corrected; *Ding and Gold, 2012a*; *Ding and Gold, 2010*). For correct trials, this convention equates the sign of stimulus coherence with the sign of the associated choice. That is, negative values map onto low-frequency stimuli and choices, whereas positive values map onto high-frequency stimuli and choices. We analyzed choice-related activity in 300 ms time bins, shifted in 10 ms steps. Choice selectivity was quantified using an ROC analysis, which measures the ability of an ROC-based ideal observer to predict a monkey's choice based only on firing rates (*Russ et al., 2008b*; *Tsunada et al., 2016*; *Tsunada et al., 2011b*).

### Linear-classifier analysis for population activity

We used linear classifiers (*Meyers et al., 2008*; *Bishop, 2006*) to test whether vlPFC population activity was modulated by stimulus coherence (using four binned ranges of coherence: [1] −100% − −50%, [2] −50 − 0%, [3] 0% − +50%, and [4]+50% − +100%) or by behavioral choice (high- versus low-frequency choices across all coherences). This analysis was restricted to data generated on correct trials only to help to ensure that we could quantify the effects of stimulus coherence and behavioral choice on vlPFC population activity and not outcome effects (correct versus incorrect trials). For each classifier and for each neuron, we z-scored firing rate and randomly subsampled the trials so that we had equal number of trials for each condition. Each classification analysis underwent a 10-fold cross-validation procedure to avoid overfitting. This procedure divided the neuronal data into 10 groups in an iterative fashion, such that one group was a test set and the remaining nine formed a training set. We implemented a linear read-out procedure in which we fit the training set to a linear hyperplane that separated the population response vectors corresponding to the two choices. For the coherence classifier, we implemented a 'one-versus-all' classification in which we built four classifiers (one for each binned coherence range) and trained each of them, in an iterative fashion, to discriminate between one particular coherence range versus all of the remaining three coherence ranges. Using the test data, we identified which of the four classifiers had the best performance and report average performance across coherence. For both classifiers, we calculated the fraction of times that the test data was classified correctly and report average performance over 1000 different instantiations of a classifier.

## Code availability

The data analyses were performed in Matlab and are available on GitHub (*Tsunada, 2019*; copy archived at https://github.com/elifesciences-publications/Joji).

## Acknowledgements

We thank Dr. Andrew Liu for help with the experimental setup, animal training, and neuronal recording. We also thank Dr. Long Ding for providing data analysis programs and for helpful discussions. Supported by R01 MH115557 to JIG, R01 DC009224 and DC012961 to YEC, and Leading Initiative for Excellent Young Researchers Grant 1071421 to JT.

## Additional information

### Competing interests

Joshua I Gold: Reviewing editor, *eLife*. The other authors declare that no competing interests exist.

### Funding

| Funder | Grant reference number | Author |
| --- | --- | --- |
| National Institute on Deafness and Other Communication Disorders | DC009224 | Yale Cohen |
| National Institute of Mental Health | MH115557 | Joshua I Gold |
| National Institute on Deafness and Other Communication Disorders | DC012961 | Yale Cohen |
| Ministry of Education, Culture, Sports, Science and Technology | Leading Initiative for Excellent Young Researchers (1071421) | Joji Tsunada |

The funders had no role in study design, data collection and interpretation, or the decision to submit the work for publication.

### Author contributions

Joji Tsunada, Conceptualization, Data curation, Software, Investigation, Visualization, Methodology, Writing—original draft, Writing—review and editing; Yale Cohen, Joshua I Gold, Conceptualization, Software, Formal analysis, Funding acquisition, Visualization, Methodology, Writing—review and editing

### Author ORCIDs

Joji Tsunada https://orcid.org/0000-0002-9125-7578
Yale Cohen https://orcid.org/0000-0002-0830-5162
Joshua I Gold https://orcid.org/0000-0002-6018-0483

### Ethics

Animal experimentation: The University of Pennsylvania Institutional Animal Care and Use Committee approved all of the experimental protocols, which were conducted under protocol 804699.

### Decision letter and Author response

Decision letter https://doi.org/10.7554/eLife.46770.012
Author response https://doi.org/10.7554/eLife.46770.013

## Additional files

### Supplementary files

• Transparent reporting form
DOI: https://doi.org/10.7554/eLife.46770.010

## Data availability

The data analyses were performed in Matlab; this code is available https://github.com/CohenAuditoryLab/Joji (copy archived at https://github.com/elifesciences-publications/Joji).

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
