## [Decision Letter]

Thank you for submitting your article "Post-decision processing in primate prefrontal cortex influences subsequent choices on an auditory decision-making task" for consideration by *eLife*. Your article has been reviewed by three peer reviewers, one of whom is a member of our Board of Reviewing Editors, and the evaluation has been overseen by Barbara Shinn-Cunningham as the Senior Editor. The following individuals involved in review of your submission have agreed to reveal their identity: Christopher I Petkov (Reviewer #2); Alexandre Pouget (Reviewer #3).

The reviewers have discussed the reviews with one another and the Reviewing Editor has drafted this decision to help you prepare a revised submission.

The reviewers all found the principal findings interesting in terms of the relationship of activity in prefrontal neurons to performance on the next trial and the effect of stimulation.

Essential revisions:

1) A main issue relates to sampling within this pair of case studies that concerns reviewer 1; reviewer 2 thinks that this can be rebutted based on the microstimulation results.

2) Reviewer 2 also suggests broadening the Discussion to consider relevance to sequence learning.

3) Reviewer 3 requires clarification of the behavioral results.

*Reviewer #1:*

The work shows delayed activity in vlPFC corresponding to sensory evidence, choice, correctness and trial success. I liked the work, which builds on previous work showing only weak decision-related activity in non-core auditory cortex. And I liked the use of manipulation with electrical stimulation.

1) The central question for me raised by the work is if these responses are not causally related to the immediately preceding choice, given they come after, then how is that encoded – dlPFC is probably where I would look first. The authors cite contradictory literature and I would agree they might just be looking at a different aspect.

2) In primate work of this sort I think it perfectly acceptable to use a group-of-case studies approach, but the two monkeys here had different strategies. I was unsure about the extent to which this qualifies the main result related to the post-decision trace in terms of arguing for a universal mechanism.

3) Another concern is also about power and inference. The frontal cortex is quite patchy with respect to responses related to perception. The authors refer to previous studies that do show PFC forms decisions and I completely agree this might reflect sampling.

Overall I think the authors are saying something important about decision correlates. I think the absence of a causal explanation for the decision, which I suspect was the thing sought, should not be over-interpreted based on the absence of any clear basis demonstrated in this sample of neurons.

*Reviewer #2:*

The study by Tsunada and colleagues is very interesting and impressive because it reports vlPFC neuronal responses from the primate brain that are related to subsequent decisions that the monkeys will make using also Drift Diffusion Modelling. The monkeys' behavioural strategies on subsequent trials are interestingly enough idiosynchratic, which provides unexpected insights on the involvement of vlPFC in decision making. The authors also use microstimulation to show that the current trial performance is not necessarily affected by microstimulation throughout the trial, but that the monkeys' decisions on the next trial are influenced. So the role of areas 45 and 46, where they find these neurons, is much more interesting than previously thought with regards to decision related processes.

1) Clearer presentation of results: Although I'm generally convinced, the presentation of results and figures didn't always seem to line up for me. There were several instances where I simply could not see what the authors were referring to in the figures, even after scrutinizing the Results text and figures several times. I was also convinced that you see stimulus driven, coherence related, decision related, lever related and juice related responses, although you seem to dismiss these and emphasize the effects related to the next trial. So I would strongly recommend that you check and revise the Results presentation, including figures, to ensure it is clear and that the claims in the text lines up with the Results figures: check how your statements are supported by the results, which will require better explication and closer reference to what you point to in specific figure panels for key claims.

2) Stave off two case study criticism: to stave off criticism that the idiosynchratic patterns are just two case studies rather than consistency in the two animals (suggesting that the patterns are consistent and likely to generalize to other monkeys regardless of the strategies they take with regards to the next trial), I would encourage you to pick this up in the Discussion and point to the data that shows consistent microstimulation effects that relate to the specific decision that the monkey will take next, again related to their strategy.

3) More complete Discussion: vlPFC has been seen, at least with neuroimaging studies in humans and monkeys, to be sensitive to within trial sequence order effects, or in humans’ language syntax. This suggests that the signals in these parts of the brain, at least in humans, are involved in constructing sequential information, typically on the order of a few seconds. The Discussion and broader appeal of your work could benefit from some consideration of how your results from this part of PFC might inform sequential processing and the time scale at which you think these operations occur, at least based on your data. The Discussion is cursory and could also benefit from considering whether the situation would be different in premotor cortex (v6) and motor cortex (M1). Presumably it is and this sort of future planning is certainly likely to involve more anterior parts of PFC, so it would be good to couch the vlPFC results within the auditory sequence/language processing literature that typically converges in terms of effects in this part of the brain, and with adjacently interconnected territory in PFC/premotor cortex. Relatedly I couldn't quite follow the mechanistic insights in how you view the vlPFC signals to coordinate with those in belt auditory cortex (field AL). So this part too could be more clearly discussed.

*Reviewer #3:*

This manuscript presents experimental work that builds upon the authors' previous work (Tsunada et al., 2016). While their previous work focused on causal contributions of two primate brain regions (AL- and ML-belts of auditory cortex) to an auditory perceptual decision, the present study investigates neural responses in a downstream region (vlPFC). The authors report that the vlPFC seems to encode primarily post-decision neural signatures for the choice, outcome, and uncertainty in the sensory evidence. Consistent with this finding, microstimulation of this area was found to affect decision on the next trial, thereby revealing causal evaluative processes for the task that affect future behavior.

Overall, I think that this paper will make a great contribution to the literature.

1) It is unclear whether there are only two frequency values, one low and one high, or whether there are multiple low- and high-frequencies. Judging by the authors' previous work (Tsunada et al., 2016) and in the subsection “Data-collection strategy” of the current manuscript, it seems like there is a range of frequencies used for the low-high task. If this is indeed the case, then in a ± 100% coherence task, how do the monkeys know whether the frequency being played is low or high? Are they demarcated by a particular frequency that they monkey has learned? Moreover, is the difference between the low- and high-frequency constant? If not, then how do the authors account for the effect this will have on the behavior?

2) The behavioral analyses and the experimentally rewarded trials seem to be at odds with each other. Based on each trial's coherence, which presumably guides reward on that trial, a tone burst sequence is generated by randomly assigning the frequency of each tone burst to the low- and high-frequency value for that trial. However, for the behavioral analyses, the stimulus coherence was calculated from the actual proportion of low- and high-frequency tone bursts that were presented from the stimulus onset until the monkey indicated its choice by moving the joystick on that trial. In other words, the task seems to be designed such that the monkeys need to infer a latent variable (coherence), which is not what the behavioral analyses seem to be doing.

To illustrate this point, consider the following. Given that tone bursts last 50ms with 10ms inter-burst intervals, only a few tone-bursts can be presented before the subjects' average RTs (about 8 bursts in 0.5 seconds). So, in a +20% (more high-frequency) coherence trial, it is possible just by chance, that there are more low-frequency bursts before choice, leading the monkey to make an incorrect decision and hence not being rewarded. However, the analyses would consider that the trial had a correct response. Please clarify.

3) The authors have not mentioned whether they pooled all neurons across trials and monkeys for the classification of choice and coherence (Figure 6), but that seems to be the case.

4) The authors may want to discuss the compatibility of their results with the following studies that look at the role of vlPFC in decision-making in monkeys:

– Baxter et al. (2009) show that lesioning vlPFC in rhesus macaques impacts a strategy-based task, but not value-based decision-making.

– Rudebeck et al. (2017) show that vlPFC is critical in probabilistic learning of stimulus to outcome (state to reward) by performing lesion studies in rhesus macaques.

---

## [Author Response]

Essential revisions:1) A main issue relates to sampling within this pair of case studies that concerns reviewer 1; reviewer 2 thinks that this can be rebutted based on the microstimulation results.

We now highlight this important point in a new paragraph of the Discussion:

“The fact that our two monkeys used different biasing strategies (win-stay for monkey T, win-switch for monkey A) affects the interpretation of our findings in two primary ways. […] This result implies that the vlPFC provides a choice-dependent signal that is used to generate sequential biases but does not participate directly in forming the idiosyncratic strategies that use those biases.”

2) Reviewer 2 also suggests broadening the Discussion to consider relevance to sequence learning.

We now end the Discussion by speculating about this interesting point:

“In addition to recording from large neuronal populations, it would be instructive to examine vlPFC activity under a broader range of task conditions to better understand its general role in the sequential processing of auditory information. […] It would be interesting to explore whether monkey vlPFC may possess a precursor system used in human language.”

3) Reviewer 3 requires clarification of the behavioral results

We have made extensive revisions throughout the main text, including the Results and Materials and methods sections, to add clarity and address the specific concerns raised by reviewers 2 and 3.

Reviewer #1:[…] 1) The central question for me raised by the work is if these responses are not causally related to the immediately preceding choice, given they come after, then how is that encoded – dlPFC is probably where I would look first. The authors cite contradictory literature and I would agree they might just be looking at a different aspect.

We agree that this is a central, unanswered question of this work, and we appreciate the interesting suggestion. We now consider the dlPFC along with other brain areas including the parabelt region of auditory cortex in the Discussion (fifth paragraph).

2) In primate work of this sort I think it perfectly acceptable to use a group-of-case studies approach, but the two monkeys here had different strategies. I was unsure about the extent to which this qualifies the main result related to the post-decision trace in terms of arguing for a universal mechanism.

Thank you for pointing out this important issue. As described above, we have added a paragraph to the Discussion to more clearly describe how we interpret our results in the context of these different strategies (including, as suggested by reviewer 2, emphasizing the importance of the microstimulation effects to our overall conclusions about the role of the vlPFC in the sequential choice biases).

3) Another concern is also about power and inference. The frontal cortex is quite patchy with respect to responses related to perception. The authors refer to previous studies that do show PFC forms decisions and I completely agree this might reflect sampling.

We fully agree and note in the Discussion that our results might reflect the fact that “we sampled a different PFC population than in these other studies” that implicated a role for the PFC in decision formation.

Overall I think the authors are saying something important about decision correlates. I think the absence of a causal explanation for the decision, which I suspect was the thing sought, should not be over-interpreted based on the absence of any clear basis demonstrated in this sample of neurons.

Again, we fully agree that our data do not allow us to make overly broad inferences about the role of the many different subregions of PFC in forming auditory decisions. We thus focus on the clear feature of our data: post-decision activity that is related to sequential choice structure.

Reviewer #2:[…] 1) Clearer presentation of results: Although I'm generally convinced, the presentation of results and figures didn't always seem to line up for me. There were several instances where I simply could not see what the authors were referring to in the figures, even after scrutinizing the Results text and figures several times.

We apologize for the confusion and have made extensive revisions throughout the Results and Materials and methods to address reviewer’s comments.

I was also convinced that you see stimulus driven, coherence related, decision related, lever related and juice related responses, although you seem to dismiss these and emphasize the effects related to the next trial. So I would strongly recommend that you check and revise the Results presentation, including figures, to ensure it is clear and that the claims in the text lines up with the Results figures: check how your statements are supported by the results, which will require better explication and closer reference to what you point to in specific figure panels for key claims.

We again apologize for the confusion and certainly did not intend to appear dismissive of our findings of stimulus-, choice-, and outcome-related modulations of neuronal activity in vlPFC. Rather, we emphasize that because these modulations occur after the decision is completed, they cannot play a role in forming that decision but instead appear to play a role in evaluating that decision (e.g., via a confidence or reward-expectation variable) that is used as part of a trial-by-trial learning process to adjust the subsequent decision. We now clarify this point in the manuscript, particularly in the Results and the first two paragraphs of the Discussion.

2) Stave off two case study criticism: to stave off criticism that the idiosynchratic patterns are just two case studies rather than consistency in the two animals (suggesting that the patterns are consistent and likely to generalize to other monkeys regardless of the strategies they take with regards to the next trial), I would encourage you to pick this up in the Discussion and point to the data that shows consistent microstimulation effects that relate to the specific decision that the monkey will take next, again related to their strategy.

Thank you for this excellent suggestion. As noted above, we address this issue directly in a new paragraph of the Discussion.

3) More complete Discussion: vlPFC has been seen, at least with neuroimaging studies in humans and monkeys, to be sensitive to within trial sequence order effects, or in humans language syntax. This suggests that the signals in these parts of the brain, at least in humans, are involved in constructing sequential information, typically on the order of a few seconds. The Discussion and broader appeal of your work could benefit from some consideration of how your results from this part of PFC might inform sequential processing and the time scale at which you think these operations occur, at least based on your data. The Discussion is cursory and could also benefit from considering whether the situation would be different in premotor cortex (v6) and motor cortex (M1). Presumably it is and this sort of future planning is certainly likely to involve more anterior parts of PFC, so it would be good to couch the vlPFC results within the auditory sequence/language processing literature that typically converges in terms of effects in this part of the brain, and with adjacently interconnected territory in PFC/premotor cortex.

Thank you for bringing up these interesting ideas, which we now speculate upon in the final paragraph of the Discussion.

Relatedly I couldn't quite follow the mechanistic insights in how you view the vlPFC signals to coordinate with those in belt auditory cortex (field AL). So this part too could be more clearly discussed.

Thank you for highlighting this important point. We indicate in the Discussion (fifth paragraph) that the monkeys’ behavioral patterns are consistent with an accumulation of the sensory evidence that in our previous study we found is represented in AL. However, in this study we did not find that vlPFC contributes to this accumulation process. We therefore propose other brain regions that might implement the accumulation, as follows:

“One possibility is a set of other brain areas that have shown information-accumulation activity on other tasks, including the dorsolateral PFC and parts of the posterior parietal cortex (PPC; Gold and Shadlen, 2007; Brody and Hanks, 2016). […] Another intriguing possibility is an auditory-specific circuit involving the parabelt region of auditory cortex, which receives direct input from AL, projects to the vlPFC, and analyzes acoustic properties of behaviorally relevant sounds (Hackett, 1999; Romanski, 1999; Petkov et al., 2004).”

Reviewer #3:[…] 1) It is unclear whether there are only two frequency values, one low and one high, or whether there are multiple low- and high-frequencies. Judging by the authors' previous work (Tsunada et al., 2016) and in the subsection “Data-collection strategy” of the current manuscript, it seems like there is a range of frequencies used for the low-high task. If this is indeed the case, then in a ± 100% coherence task, how do the monkeys know whether the frequency being played is low or high? Are they demarcated by a particular frequency that they monkey has learned? Moreover, is the difference between the low- and high-frequency constant? If not, then how do the authors account for the effect this will have on the behavior?

We appreciate the careful reading and now provide additional details in Materials and methods:

“During testing, we generally used 1250 and 2500 Hz as low and high frequencies, respectively (*n*=12 out of 29 sessions for monkey T, 17 out of 39 sessions for monkey A). Otherwise, we used other values, with low/high values always less/greater than 1750 Hz, and with the two values in a given session always separated by 1–3 octaves.”

Also, we now plot the results of the behavioral logistic analysis in Figure 2D separately for the low- and high-frequency stimulus values used in a given session (1250/2500 as circles; other combinations as squares). We note in the figure legend that “that the two conditions corresponded to differences in lapse rates for monkey T but little effect on the other model parameters”.

2) The behavioral analyses and the experimentally rewarded trials seem to be at odds with each other. Based on each trial's coherence, which presumably guides reward on that trial, a tone burst sequence is generated by randomly assigning the frequency of each tone burst to the low- and high-frequency value for that trial. However, for the behavioral analyses, the stimulus coherence was calculated from the actual proportion of low- and high-frequency tone bursts that were presented from the stimulus onset until the monkey indicated its choice by moving the joystick on that trial. In other words, the task seems to be designed such that the monkeys need to infer a latent variable (coherence), which is not what the behavioral analyses seem to be doing.To illustrate this point, consider the following. Given that tone bursts last 50ms with 10ms inter-burst intervals, only a few tone-bursts can be presented before the subjects' average RTs (about 8 bursts in 0.5 seconds). So, in a +20% (more high-frequency) coherence trial, it is possible just by chance, that there are more low-frequency bursts before choice, leading the monkey to make an incorrect decision and hence not being rewarded. However, the analyses would consider that the trial had a correct response. Please clarify.

Thank you very much for your careful assessment of the behavioral analysis. Most trials used absolute coherence values >20% (84% of all trials from both monkeys), for which the sign of the actual and intended coherences were always the same. However, as the reviewer pointed out, there can be a small fraction of low-coherence trials that did not match for the sign. Because these low-coherence trials tended to be very difficult for the monkeys, they were rewarded at random, as we indicate in Materials and methods: “For trials with ambiguous stimuli (between -20% and +20% coherence), the monkey was rewarded on 50% of randomly selected trials, independent of the behavioral report.”

Further, please note that our analysis of selectivity for correct/error trials (Figure 5) used only trials with absolute coherence values >20%, for which the sign of actual and intended coherence was always the same.

Finally, we have clarified that our sequential analyses were based on whether or not the previous trial was rewarded, which was unambiguous.

3) The authors have not mentioned whether they pooled all neurons across trials and monkeys for the classification of choice and coherence (Figure 6), but that seems to be the case.

Yes, we pooled activity for all neurons across trials and monkeys for the classifier analysis. We added this key piece of information to the main text: “…across the population of 103 neurons recorded from all sessions from both monkeys for a linear classifier to decode both stimulus coherence and choice.”

4) The authors may want to discuss the compatibility of their results with the following studies that look at the role of vlPFC in decision-making in monkeys:– Baxter et al. (2009) show that lesioning vlPFC in rhesus macaques impacts a strategy-based task, but not value-based decision-making.– Rudebeck et al. (2017) show that vlPFC is critical in probabilistic learning of stimulus to outcome (state to reward) by performing lesion studies in rhesus macaques.

Thank you very much for these suggestions. We now include these references in our discussion of our findings related to sequential effects (Discussion, fourth paragraph).